# The Impact of Neglected Tropical Diseases (NTDs) on Women’s Health and Wellbeing in Sub-Saharan Africa (SSA): A Case Study of Kenya

**DOI:** 10.3390/ijerph18042180

**Published:** 2021-02-23

**Authors:** Elizabeth A. Ochola, Susan J. Elliott, Diana M. S. Karanja

**Affiliations:** 1Department of Geography and Environmental Management, University of Waterloo, 200 University Avenue West, Waterloo, ON N2L 3G1, Canada; eochola@uwaterloo.ca; 2Centre for Global Health Research, Kenya Medical Research Institute, P. O. Box 1578-40100 Kisumu, Kenya; diana@cohesu.org

**Keywords:** Neglected Tropical Diseases (NTDs), women’s health, wellbeing, sub-Saharan Africa (SSA)

## Abstract

Neglected Tropical Diseases (NTDs) trap individuals in a cycle of poverty through their devastating effects on health, wellbeing and social–economic capabilities that extend to other axes of inequity such as gender and/or ethnicity. Despite NTDs being regarded as equity tracers, little attention has been paid toward gender dynamics and relationships for gender-equitable access to NTD programs in sub-Saharan Africa (SSA). This paper examines the impact of NTDs on women’s health and wellbeing in SSA using Kenya as a case study. This research is part of a larger research program designed to examine the impact of NTDs on the health and wellbeing of populations in Kenya. Thematic analysis of key informants’ interviews (*n* = 21) and focus groups (*n* = 5) reveals first that NTDs disproportionately affect women and girls due to their assigned gender roles and responsibilities. Second, women face financial and time constraints when accessing health care due to diminished economic power and autonomy. Third, women suffer more from the related social consequences of NTDs (that is, stigma, discrimination and/or abandonment), which affects their health-seeking behavior. As such, we strongly suggest a gender lens when addressing NTD specific exposure, socio-economic inequities, and other gender dynamics that may hinder the successful delivery of NTD programs at the local and national levels.

## 1. Introduction

Cumulatively, Neglected Tropical Diseases (NTDs) affect more than one billion people globally, disproportionately those already marginalized and/or impoverished. The burden of NTDs is greatest in Low/Middle-Income Countries (LMICs) [1] due to poverty and lack of access to health and social protection systems [2,3]. Economists suggest a positive relationship between rising inequality and the fragility of economic policies since inequality undermines progress in health and education, causes political–economic instability, and undercuts social consensus, resulting in short-lived growth [4]. In sub-Saharan Africa (SSA), inequalities manifest in various parameters such as wealth, gender, and ethnicity, which may lead to exclusion from elements that make for a good life, such as education and health care [5]. NTDs trap individuals in a cycle of poverty due to their devastating effects on health, wellbeing, and social–economic capabilities [6], extending to other axes of inequity such as gender, ethnicity, and disability [7]. As a result, NTDs are most prevalent among populations least able to demand essential services such as women, children, ethnic minorities, and displaced people. Furthermore, their effects are felt in every sector, including health, agriculture, infrastructure, and education [8]. This research is part of a larger research program investigating the relationships between health, inequalities, and wellbeing in low/middle-income countries [9,10,11,12,13,14,15].

The World Health Organization (WHO) recognizes that global NTD programs have alleviated human and economic suffering among the world’s most impoverished communities. As such, the WHO suggests that NTDs may be regarded as equity tracers when identifying disparities in the provision of Universal Health Coverage (UHC) and access to health [8]. Consequently, Theobald et al. [16] find that NTD programs may play a role in promoting gender-equitable societies since gender equity shapes poverty and NTD experience in a myriad of ways including in the access of basic services. Despite this realization, NTDs continue to disproportionately affect women and girls [17], with little attention being paid toward gender equitable access to NTD interventions, particularly in sub-Saharan Africa [16].

Gender refers to the roles, behaviors, practices, and attributes that society bestows upon men and women. These socially constructed roles and practices affect various dimensions of everyday life, including health and wellbeing [18,19]. In this case, a gendered difference is not a significant problem, but the values placed on the various roles and responsibilities manifest as inequities between the genders [20]. Gender classifications change with time, space, and need in society [21]. Moreover, gender roles and relations intersect with other axes of inequity such as age, ethnicity, disability, and socio-economic status [7]. As such, gender roles and relations determine how people experience NTDs, their vulnerability to the diseases, and their care-seeking behaviors [20].

In many LIMCs, certain gender-based factors support a wide range of social policies and practices biased against women. For example, domestic and sexual violence, child marriage, and labor limit developmental opportunities for women [22,23,24], making them live in fear, compromising how well they take care of themselves, their children, and access health-care services. Furthermore, gender roles, responsibilities, and relations influence how individuals interact in society and affect the access and distribution of resources. This paper examines gender dynamics and relationships in NTD transmission cycles.

While NTDs have significant health implications for both males and females [24], females are at a higher risk of acquiring certain NTDs due to biological differences, gender roles, and family dynamics [25]. For example, both men and women suffer equally from schistosomiasis and soil-transmitted helminths (Table 1). Still, women experience more negative health effects such as iron deficiency when pregnant due to compromised immunity [26]. Similarly, Female Genital Schistosomiasis (FGS) is prevalent in SSA among adolescent girls and women; the disease presents with lesions around the vagina, bleeding, unpleasant discharge from the vagina, and discomfort during sex [27,28]. If left untreated, it may cause infertility, miscarriage, and increased susceptibility to HIV and Human Papillomavirus (HPV) [28,29], which has far-reaching economic and social consequences for both livelihoods and wellbeing.

In sub-Saharan Africa, women are caregivers in the family; this predisposes them to NTDs such as trachoma as they risk transferring bacteria into their own eyes when tending to sick family members. Women are also two to three times more likely to be permanently blinded from trachoma than men [32,33] due to poor health-seeking behavior. When a woman becomes blind, her educational opportunities and ability to earn a living become severely limited, promoting poverty [34]. Similarly, women tend to be at risk of acquiring NTDs such as Human African Trypanosomiasis (HAT) and leishmaniasis due to heightened occupational exposure to the responsible disease-carrying vector compared to men [34]. Women are also more likely to be stigmatized after contracting NTD infections compared to men. For example, women presenting with lesions from Cutaneous Leishmaniasis (CL) are more stigmatized and deemed unfit for marriage or raising children. The resulting stigmatization causes or escalates psychological disorders and restricts social participation [35]. Comparably, Person et al. [36] report that stigma affects women’s occupational roles, resulting in a loss of income, poor access to resources, decreased social interactions, and diminished social identity. Furthermore, stigmatized women experience a loss of purpose in their lives and a low self-esteem. Women who are stigmatized and labeled tend to shy away from seeking treatment and consequently suffer from complications such as infertility and death [26].

The Sustainable Development Goals (SDGs) have a significant focus on equity [37], with SDG 5 aiming to achieve gender equality and empowerment for all girls and women. Kenya, which lies in the eastern part of Africa, has a population of approximately 48 million [38] and ranks 125th out of 157 in the achievement of the Sustainable Development Goals (SDGs) [39]. In Kenya, girls and women face multifaceted problems related to gender (e.g., cultural acceptance of gender-based violence) and remain underrepresented in decision-making processes at various levels despite their unlimited potential [40]. Currently, Kenya’s economy is on an upward trend with the devolution of government offices bringing services closer to Kenyans. However, the benefits of growth remain disproportionately shared within the population, with women remaining politically, economically, and socially disadvantaged [21]. The recently promulgated constitution provides a framework for addressing gender equity [21] and women’s rights. It examines the factors that traditionally exclude women, hindering them from full involvement in growth and development. Thus, this paper aims to establish the impact of NTDs on women’s health and wellbeing in SSA using Kenya as a case study.

## 2. Materials and Methods

### 2.1. Study Area

Currently, 20% of Kenya’s land is arable, with agriculture being the main economic activity. Practices such as irrigation, fishing [41], and domestic chores are considered risk factors for the spread of NTDs such as schistosomiasis. Girls and women are also at a higher exposure rate for NTDs than boys and men due to household chores [25,41] that limit school attendance, work opportunities, and land ownership [21].

This research was undertaken in five Kenyan counties endemic for more than one NTD (Busia, Kilifi, Nairobi, Kisumu, Turkana) to explore how NTDs affect the health and wellbeing of women. The use of qualitative methods allowed for the “silenced voices” of women to be heard and enabled a better understanding of the discourses around gender inequities in the context of NTDs. Our research protocol was reviewed and approved by the University of Waterloo Research Ethics Committee (ORE# 22493) and Maseno University (Kenya) Ethical Review Board (MSU/DRP/MUERC/00496/17).

### 2.2. Data Collection

Our research was framed within social constructionism. We used purposive sampling to identify participants infected or affected by NTDs, aged 18 years and above, both male and female. Our research took on a humanistic approach to capture the lived experiences of women infected and affected by NTDs in Kenya. Our data collection took place from December 2017 to February 2018. Prior to the data collection, a reconnaissance trip was made by the lead author (who was a female Ph.D. student at the time) in December of 2016 for approximately one month. The purpose of the reconnaissance trip was to understand the study context and meet with the key stakeholders involved in NTD activities in Kenya. The trip to Kenya provided an opportunity to connect with individuals from each of the counties who would act as primary contacts and assist with the recruitment of research assistants. The trip was an important process as it facilitated discussions around field logistics. In December of 2017, just before the data collection commenced, the student researcher met with the county officials and community leaders from the five counties to introduce the study and seek formal permission to carry out the research within the community. Following the stakeholders meeting, the student researcher, together with the community leaders, held a forum with the community members to discuss the general purpose and objectives of the research and to address any questions and concerns the community may have had regarding the study. A total of 23 individuals were initially approached for the Key Informant Interviews (KIIs); two of them declined the interviews due to time constraints. Hence, 21 KIIs were conducted with diverse participants from the national level, including NTD managers, local NTD partners, and policymakers (Table 2). The respondents were approached by the researchers through email and telephone calls to book appointments. The use of KIIs provided insight on specific aspects of NTDs, gender, and policy developments that could influence future practice which were not easily obtained by other forms of data collection.

An interview guide (Table A1) was pilot tested that listed all the topics and issues of interest for the discussions. Subtle probes were applied to get detailed information from the participants. Interviews were carried out in either English/Swahili, which lasted between 30 and 90 min. There were no repeat interviews, and the discussions were concluded when no new information emerged (saturation). The use of theoretical saturation in the data collection process allowed for the examination of the main variation of the phenomenon identified and which could potentially fit within or fall outside the proposed theoretical constructs. In other words, the saturation process allowed for a maximum variation in sampling while at the same time maintaining an openness for themes to emerge [42]. Furthermore, through saturation, it became easier to identify confounding cases which made it easier to explore the nature of the differences. During the field research, notes were taken by the research assistants to supplement the audio recordings, and all voice data were transcribed verbatim. Pseudonyms were used to protect the participants’ privacy.

A total of five Focus Group Discussions (FGDs) of 7–8 participants each (Table 3) were undertaken, ranging in length from 45 to 90 min. There were no repeat interviews, and the discussions were concluded when no new information emerged (saturation). The FGDs differed from the KIIs, in that research participants were interviewed in a group setting, which allowed them to engage in collective dialogue. Furthermore, FGDs provided an opportunity for the participants to freely discuss their daily challenges and experiences in a supportive environment. Most focus group discussions were conducted in Swahili because the study populations were heterogeneous. The use of Swahili also ensured that the participants were comfortable throughout the interviews, which limited the power dynamics between them and the female researcher. A structured FGD guide provided a general overview of the topics during the discussions with the flexibility to probe for additional information. All the voice data from the FGDs were captured on audio recorders and transcribed verbatim.

As stipulated, qualitative research calls for researchers to understand their positionality and remain reflexive for a rigorous outcome [43]. Researchers are required to be aware of their connection to the research by examining the power dynamics between them and the study participants which may manifest through previous knowledge of the study area, perceptions, and embodying emotions [43]. In our case, the lead author (female, Ph.D. candidate) who conducted all the KIIs and FGDs was born and brought up in Kenya; there was a possibility that her preconceived knowledge of the country could have potentially affected the research process. To minimize such biases, we relied on evidence from the existing literature to formulate open-ended questions that allowed the participants to answer questions based on their perspective with no right or wrong answer; hence, all participants’ voices were heard. The research team also engaged in member checking during the interviews where participants were asked to verify the researcher’s summary of the content provided during the discussions. Member checking helped address any misinterpretation that may have occurred and ensured an accurate presentation of the results.

During thematic analysis of the data, transcripts were scanned to generate a coding manual. The codes were developed using a deductive approach that captured themes that corresponded to the interview questions, existing literature, and concepts, while inductive codes considered issues emerging from the transcripts. The data was coded line-by-line to produce textual elements, which were later organized into themes and sub-themes. The organization of the data enabled the determination of the similarities and differences, and it ensured proper comparison and connection of the themes to the research objectives. Two transcripts were independently coded for each data source, first by the lead researcher and secondly by an independent researcher, to establish the inter-rater reliability. The aim was to determine at least a 70% agreement for each source, as described by Miles and Huberman [44].

During data collection, we minimized bias by adhering to data collection schedules and refining the interview guides. We also ensured that the transcription process was accurate with an efficient coding of themes to enhance the analysis and interpretation of the data. Through triangulation of methods (KIIs and FGDs), we ensured that the data was dependable and conformed to the respondents’ thoughts and lived experiences.

## 3. Results

### 3.1. Gender-Specific NTD Exposure

The women and girls who participated in this research had assigned gender roles and responsibilities, which they were expected to perform, e.g., caring for the family, cooking, cleaning, washing clothes, and fetching water. The results demonstrated that men residing in these areas expected women to carry out the duties without compromise, earning some men the title of “male chauvinists”. Due to heavy responsibilities being placed on women in such traditional settings, most women and girls were at risk of waterborne NTDs such as schistosomiasis, as noted explicitly by the KI below:

*You see the male chauvinists […] they believe that women should take up some roles […] it is like the women are the ones go to the rivers to fetch water, wash the clothes… which has some impact on schistosomiasis infection.*.(Antony, male 47 years, KI)

We also found that most NTDs have health-related impacts in the short and long term. For example, when left untreated, schistosomiasis presented gender-specific complications in females that manifested in the form of Female Genital Schistosomiasis (FGS), affecting the reproductive organs as stated by the FG discussant below:


*Most of the NTDs will generally have such impacts in the long term in women and girls when left unchecked […] for schistosomiasis, you find girls getting female genital schistosomiasis and even the removal of the uterus and things like that.*
(Jane, female 51 years, FGD)

Our analysis confirmed that women were the sole caregivers in most communities. However, in Turkana county, women performed additional duties, such as building houses, rearing animals, and caring for children in an environment that had limited access to water and sanitation, among other resources. As such, women were highly exposed to trachoma as they were unable to access clean water to wash hands and prevent contamination:

*Women in Turkana [county] are at risk of more infections compared to other people, mainly because of our duties. We are the ones who care for the children, so in the process of caring for the children in a place that has a scarcity of water, you find that we are caring for our children in poor sanitation […] with the poor sanitation, we do not wash our hands […] we contract diseases like trachoma [NTD]*.(Anne, female 34 years, FGD)

Both the KI and FG participants concurred that women take on too many responsibilities around the home under unhygienic conditions that expose them to NTDs.

*When you come to the chores that are done in the family, you find some people, in my view, are overburdened, which puts them at risk compared to others. For instance, women in Turkana [county] are usually at risk of more infections than other people, mainly because of the duties that they perform. They are the ones who are willing to care for the children, so in the process of caring for the children […] where there is a scarcity of water, you find that they are caring for their children under poor sanitation, with the poor sanitation, one is even able to remove excreta by bare hands and may not even wash their hands. […] you understand what happens there in terms of contracting diseases like trachoma [NTD]*.(Mary, female, 40 years KI)

We also found that due to caregiving duties, women were obliged to care for ill children suffering from trachoma. Trachoma is caused by Chlamydia trachomatis, which is a highly infectious bacterium and a leading cause of active trachoma in children.

*In trachoma, 2/3 s of those affected are female, and 1/3 are men, the reason being children are the ones affected by active trachoma, especially the age 1–9 and women being the caretakers of children in our society they tend to spend a lot of time with the children, attending to them, wiping them, getting medication to them all the time, contaminating their fingers, so in the process, they contaminate their own eyes and get infections*.(Benjamin, male, 33 years KI)

Women also carried out duties that required them to step out of the homestead such as fetching water, firewood, and building materials, which expose them to a range of disease-carrying vectors such as sandflies that transmit visceral leishmaniasis (Kalazar):


*Women may not go far from home, but sometimes when they go to look for firewood where transmission is taking place, they may be bitten by sandflies*
(Grace, female, 27 years, FGD)

### 3.2. Financial and Time Constraints to Health-Care Access

Results indicated that women lacked access to health-care services due to financial and time constraints. In certain circumstances, women were required to seek their husband’s permission before accessing treatment or even taking part in communal NTD interventions within their community. Most community members interviewed felt that patriarchy influenced decision making and accelerated disease progression:

*Sometimes your husband will say, ‘No, you cannot be able to attend the clinic,’ and that is the only day there is outreach in the community. So you have to wait for another time, and the disease progresses, the scarring of the tissues of the eye progresses until now it gets to the cornea and cannot be reversed, so you see there is that effect of the gender relations*.(Vanessa, female 33 years, KI)

We also found that women infected with trachoma, in addition to seeking permission from their husbands to access health care required physical and financial support from their spouses in instances where they had been blinded by an advanced stage trachoma infection:

*[…] sometimes, you find that a lady has trachoma, sometimes you have to get permission from your husband to be able to access health care. Or sometimes, you could be having a need for financial and physical support since you are blind; you need someone to guide you to the nearest facility*.(Maria, female, 27 years, FGD)

Most of the NTDs present in the study populations had long-term effects; for example, a bite from a snake may not kill, but the venom injection in the body may have prolonged consequences on organs such as the kidney and slowly lead to death. Women who had children disabled or disfigured from snake bites found that they had to abandon any form of paid work to care for their children, as stated by the KI below:

*We are talking about the chronic nature of the condition […]. In snakebite, you are bitten; the effects are long term. Kidney problems, if not attended to in time, also result in death, and in all this, the mother has to stop what they are doing to attend to this kid*.(John, male 39 years, KI)

Our analysis from the KI and FG discussants confirmed that NTD infection is devastating to families who are poor and possess limited funds. In particular, we found that women were overburdened with household activities around the home, such as taking care of the children and the livestock, which presented substantial risk factors for the transmission of NTDs such as trachoma. Once a woman became infected, their income earning potential became compromised:

*I know that most of the time, if a lady is taking care of the household as the head and is also taking care of the livestock when they get their eyes infected, it reduces their opportunity of going out to the markets to sell their produce*.(Boaz, male 42 years, KI)

In addition, in cases where a child or a family member became infected with an NTD, and they are referred to a hospital out of town, women are obliged to drop everything and accompany the person seeking medical attention until a time when they recovered and were discharged from the inpatient hospital stays: *When they say the child is being referred to Eldoret [town] or whichever hospital, the mother accepts to go with the child and stay with the child until they are well* (Aisha, female 38 years, FGD).

### 3.3. Desertion, Stigma, and Collapse of the Family Structure

This analysis revealed that the social challenges associated with NTD infection included desertion, stigma, and discrimination, which were also linked to incidences of abandonment, divorce, and the collapse of family structure. Results indicated that women infected with NTDs faced more stigma and discrimination from the community compared to men. When they required financial support to go to the hospital, they were not taken seriously. Instead, they were forced to stay at home for prolonged periods, which made them miss out on work and an opportunity to earn a living as stated below:

*I can say there is stigma and discrimination because when I say I am unwell as a woman; people do not take me seriously. When I say I need money to go to the hospital, they take it lightly. I cannot work, and this forces me to stay home even for a whole month, so I see a lot of discrimination*.(Aisha, female 37 years, FGD)

We found that girls affected by NTDs that maim, such as snake bites, were equally stigmatized, socially discriminated against, and deemed less acceptable. Results showed that such girls found it challenging to be productive and earn an income, trapping them in a cycle of poverty. In the future, they also found it challenging to find a spouse:

*The stigma in snakebite. You are bitten, you get maimed, yeah, if you are not dead, then you are assured that there will be some stigma associated with that. […] I gave you an example of the girl child; yeah, we have kids, and you know how girls look at their beauty. You are bitten by a snake; socially, there is the feeling that you are less acceptable even when getting a spouse in the future. In terms of also just getting a source of livelihood as a social factor, if you are bitten and now maimed in one way or another, then you cannot be as productive as someone who is fully physically able. This reduces your income somehow. Then at the end of the day, you remain within poverty*.(Leonard, male 22 years, FGD)

These data suggest that infection with NTDs interfered with a woman’s capacity to relate intimately with her husband, became a source of conflict, and contributed to the breakdown of the marriage:

*These diseases really affect us even when you look at the domestic [sexual] relations it is really affected because it slows down and this interferes with the marriage, and the man will leave you*.(Grace, female 22 years, FGD)

Interestingly, some of our respondents noted that women were more compassionate when men were infected with NTDs; women showed sympathy and compassion to their partners, but when women were infected, the majority of the men abandoned them to suffer:

*In the community where I am working, if a man is affected and infected, the woman has more passion and sympathy. But when it comes to a woman, men, in many cases, they do not care. They leave the house and come back late. Women tend to have more sympathy than men. So, if the NTD affects a woman, the woman suffers more*.(Brian, male 43 years, KI)

## 4. Discussion

This research aimed to establish the impact of NTDs on women’s health and wellbeing in SSA with Kenya as a case study. Findings revealed that NTDs disproportionately affect women and girls due to their assigned gender roles and responsibilities; women take on heavy duties such as taking care of the home and family, which places them at risk of NTDs such as schistosomiasis, trachoma, and kalazar. If left untreated, such NTDs present long-term health impacts of infertility, blindness, and organ damage, respectively. We also find that as women care for their children infected by NTDs, over time, they risk acquiring the same NTDs through contamination, especially for trachoma cases, as a result of a direct link between trachoma infection and limited access to water and sanitation facilities.

Women face financial and time constraints when accessing health care, since most women engage in unpaid work, which diminishes their economic power and autonomy. For women who engage in employment, when a household member is infected by an NTD, especially children, they are obligated to give up their jobs and care for the person however long it takes, which compromises their income-earning capabilities. Furthermore, in heavily patriarchal communities of Turkana County, women may find themselves in situations where they have to seek permission from their husbands to access NTD health services or even participate in NTD interventions within the community. Such, limited support for health-care services accelerates disease progression, leading to disability, disfigurement, and subsequently poor health and wellbeing.

We found that NTD infection presents social challenges such as stigma, discrimination, abandonment, and a collapse of the family unit. Our results reveal that women suffer more from the social consequences of NTDs compared to men, which affects their health-seeking behavior. Due to stigma and discrimination, women encounter constant marital challenges, which leads to the collapse of the family union. Furthermore, women who are physically disabled by NTDs such as trachoma and snake bites find it challenging to find a spouse. In cases where the women are married, they risk being deserted by their spouses, but on the contrary, women show sympathy and compassion when men are infected by NTDs.

In the broader context, these results confirm that gender inequity propagates gender-specific exposure to NTDs due to poverty and an unequal distribution of (economic and social) power. Angus Deaton [45] affirms that when power is distributed unequally, it results in high poverty rates, lack of access to health care, gender inequality, illiteracy, and conflict. Similarly, Tomaney [46] finds that people living in poor communities and suffering from the socio-economic impacts of infectious diseases such as NTDs demonstrate poor health-seeking behavior and wellbeing across age and gender. Our results establish that women face financial and time constraints when accessing NTD services, which accelerate disease progression; consistent with this observation, women make up the vast majority of the world’s poor and live in extreme poverty [47,48], unable to afford basic services. As such, when a woman lacks the economic resources, education, political and social influence and is struggling to feed her family, she misses to accord NTD infections the urgency required when her health and that of her family is affected.

We established a link between desertion, stigma, and collapse of family structure and find that stigma fuels both disease burden and poverty levels. Similarly, we find that women have a higher disease burden from NTDs than men due to stigma, desertion, and reduced health care access [17,36,49]. In such cases, women report they are not taken seriously and are forced to remain at home for prolonged periods without an income. Consistent with our findings of stigma and discrimination, women suffering from the chronic effects of lymphedema and elephantiasis report finding it difficult to find a spouse because of the swelling of the extremities, which interferes with mobility and the inability to participate in social activities, all leading to hopelessness [26,50]. For this reason, Manderson et al. [51] suggest that women need to be empowered to access treatment to enhance their health and wellbeing as well as prevent them from being disease reservoirs for NTD transmission in their communities [52]. As it is, women are disproportionately affected by poverty and illiteracy; they lack land rights and have little political voice, which increases their exposure to NTDs. As such, Bangert et al. [53] advocate for women empowerment as a key aspect of NTD control as it offers an opportunity to improve the health and rights of women and girls in affected populations [34].

In the family context, we found that women suffering from disabling NTDs such as trachoma and lymphatic filariasis are barred from engaging in sexual relations, which causes them to experience stigma as well as social and psychological distress. Similarly, women involved in a marital conflict are forced to leave their matrimonial homes, while males seek sexual relations and companionship from other women [54].

In Kenya, culture influences human activities and the community’s perception of the interaction between disease and the environment. Moreover, culture determines social behavior, occupation, and gender-specific knowledge on the impact of the interactions between hosts, parasites, and vectors [55]. In our analysis, culture was not a key driver of NTD infections, but it was an underlying factor for gender-specific exposure to NTDs, stigma, and a determinant in health-seeking behavior. van Brakel [56] argues that despite the cultural or religious diversity across African countries, the experience of stigma remains consistent across settings as stigma affects mobility, relationships, marriage, employment, and participation in leisure activities, religious, and social functions. Furthermore, the magnitude, depth, and experience of stigma are linked to gender roles, relations, and expectations [53,56]. Similarly, Vlassoff et al. [57] examine the gender-related features of stigma due to NTDs, such as lymphatic filariasis, onchocerciasis, and leprosy. The results indicate a gendered difference in the manifestations of stigma; in men, stigma is perceived as a limitation to economic opportunities, while in women, stigma is viewed as a barrier to marriage and family [35,57].

We are currently in the midst of a significant infectious disease epidemic, which has led to considerable overcrowding in hospitals and put a strain on all health facilities in Kenya. Even though men’s mortality rates are projected to be higher during the COVID-19 pandemic, women continue to suffer from the pre-existing socio-economic inequalities, which have worsened as women hold less secure jobs, earn less, and have diminished access to social protection. As such, the impact of NTDs on women has increased, and if the current trend continues, it could reverse the limited progress made toward gender equality and women’s rights [58].

Our research used qualitative methods to highlight the lived-in experiences of NTDs on women’s health in the primarily biomedical field of infectious diseases. In a study such as this, there were limitations that are to be considered when interpreting the results. First, our research was qualitative and based on self-reported data, which is susceptible to small sample sizes and social desirability. In this case, participants could overstate or understate their experiences, depending on expectations. To minimize research bias, we asked follow-up questions and used subtle probes across our participants (KIs and FGDs) to enhance the experience. Second, our study was cross-sectional, which means we collected data at a single time point for the KIs who were interviewed alone and for the focus group discussants interviewed in a group; hence, our study design did not allow for the examination of potential changes over time. However, with the triangulation of methods, we were able to minimize bias and ensure similarity in the emerging themes between the FGDs and the KIIs. Third, in as much as our results demonstrated the reality of NTDs on women’s health and wellbeing, we acknowledge that a future use of quantitative data may provide a complementary perspective on NTD prevalence at the county and country level. Additionally, we project that the use of quantitative data may enhance the interpretation of the spatial distribution of NTDs in Kenya. Thus, our research proposes the use of mixed methods in subsequent research to provide an opportunity to compare multiple perspectives and triangulate the observed research findings.

## 5. Conclusions

Overall, these findings reveal a gender inequity in the causes and consequences of NTD infection in Kenya. As a result of gender inequity, women have limited access to safe water and sanitation systems, health services, economic, and social support systems despite being the primary caregivers for their families and having related obligations to contribute to their communities.

Based on these findings, we suggest a gender equity lens in understanding and addressing the impact of NTDs on health and wellbeing in sub-Saharan Africa. A gender equity lens promotes an equivalence in life outcomes for men and women based on their different needs and interests with an overall desire to redistribute power and resources. As such, a gender lens will help address gender-specific NTD exposure, causes of socio-economic inequities, and other gender dynamics that hinder the successful delivery of NTD programs at the local and national levels. Secondly, we recommend community empowerment programs to educate and strengthen women’s autonomy on their rights to seeking health care. This will go a long way in addressing Sustainable Development Goal #5—women’s empowerment—as that intersects with Sustainable Development Goal #3—health and wellbeing for all.

## Figures and Tables

**Table 1 ijerph-18-02180-t001:** A table showing the Neglected Tropical Diseases (NTDs) of priority as per the World Health Organization (WHO).

Disease/Condition	Alternative Name	Prevalence	Pathophysiology	Causative Agent	Suspected/Found in Kenya
American trypanosomiasis	Chagas disease	0%	Cardiomyopathy	Protozoa	No
Buruli ulcer	Bairnsdale ulcer	No current data	Necrotizing skin lesions	Bacteria	Yes
Chikungunya		10–20%	Debilitating joint pain and swelling	Virus	Yes
Cystic echinococcosis	Hydatidosis	<5.3%	Depending on cysts size and location. May cause damage to the liver, lungs and brain	Helminth	Yes
Dengue		<10%	Platelet depletion and hemorrhage leading to death due to Hemorrhagic dengue fever	Virus	Yes
Endemic treponematoses	Yaws/endemic syphilis	0%	Chronic disfigurement of skin, bone, and cartilage	Bacteria	No
Foodborne trematodiases		No current data	Severe liver and lung disease	Helminths	Yes
Human African trypanosomiasis	Sleeping sickness	5–8%	Neurologic problems and death	Protozoa	Yes
Leishmaniasis		15–35%	Enlarged spleen and liver	Protozoa	Yes
Leprosy	Hansen disease	<5%	Nerve damage and limb disfigurement	Bacteria	Yes
Lymphatic filariasis	Elephantiasis	<5%	Extreme swelling of the limbs	Helminth	Yes
Mycetoma and deep mycoses	Madura foot	No current data	Destruction, deformity, and loss of tissue function	Fungal/Bacterial	Yes
Onchocerciasis	River blindness	No current data	Intense itching, rashes, and blindness	Helminth	Yes
Rabies		7–10%	Severe inflammation of the brain spinal cord, death	Virus	Yes
Scabies and other ectoparasites		<10%	Septicemia, heart disease, and chronic kidney disease	Parasitic	Yes
Schistosomiasis	Snail fever	17–30%	Enlargement of the liver and spleen	Helminth	Yes
Snakebite envenoming		1–10%	Kidney failure, tissue death, and breathing problems		Yes
Soil-transmitted helminths	Worms, STHs	12–15%	Rectal prolapse, blood and protein loss	Helminths	Yes
Taeniasis/cysticercosis		No current data	Seizures	Helminth	Yes
Trachoma		11–28%	Damage to the cornea and irreversible blindness	Bacteria	Yes

Modified from: World Health Organization [30] (see Mitra and Mawson [31] for further reading).

**Table 2 ijerph-18-02180-t002:** Characteristics: key informants.

Name	Male	Female	Number Recruited (Total 21)
Gender	16	5	21
Role in the community			
Policymaker	11	1	12
Community leader	2	1	3
Researcher	2	1	3
Non-Governmental organizations (NGO)	1	2	3
Age			
18–35 years	5	1	6
35–60 years	9	4	13
Over 61 years	1	1	2
Years of work			
1–5 years	12	3	15
5–10 years	4	2	6

**Table 3 ijerph-18-02180-t003:** Sample characteristics: focus group participants.

Name	Male	Female	Number Recruited (Total 46)
Gender	25	21	46
Place of birth			
Born in the community	19	13	32
Born outside the community	8	6	14
Length of stay in the community			
Less than 5 years	6	4	10
5–10 years	7	5	12
More than 10 years	20	4	24
Economic activities			
Casual Laborer	2	4	6
Salaried workers in the county	1	1	2
Small-scale farmers	4	7	11
Small business operator	5	8	13
Unemployed	4	10	14

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
