# Peer review of "The Impact of Neglected Tropical Diseases (NTDs) on Women’s Health and Wellbeing in Sub-Saharan Africa (SSA): A Case Study of Kenya"

_ijerph, 2021, doi:10.3390/ijerph18042180_

Round 1
Reviewer 1 Report
Thank you for submitting this interesting manuscript.
The topic and rationale for the topic are explained appropriately. I note that this is part of a larger study and this could have been made more explicit within the paper.
In the conclusion you comment that this study has used a lived experience approach to the research and this could have been described more clearly within the methodology section. This would assist with connecting the methods of data collection to the data analysis. It could have been explained more fully why KIs were needed as well as FGDs. Is there an expectation that KIs will be able to influence future practice or development? Some comment on who undertook the interviews and FGDs would have been useful and whether this would have any impact on the discussions.
The findings are presented appropriately and are sufficiently illustrated with participant voice. The discussion appears relevant to the findings presented.
Author Response
Reviewer 1
Thank you for submitting this interesting manuscript.
- The topic and rationale for the topic are explained appropriately. I note that this is part of a larger study and this could have been made more explicit within the paper.
We have highlighted this suggestion in the abstract section-line 17-19.
- In the conclusion, you comment that this study has used a lived experience approach to the research, and this could have been described more clearly within the methodology section. This would assist with connecting the methods of data collection to the data analysis. It could have been explained more fully why KIs were needed as well as FGDs. Is there an expectation that KIs will be able to influence future practice or development? Some comments on who undertook the interviews and FGDs would have been useful and whether this would have any impact on the discussions.
We have explained this recommendation in line 131-132, 149-150, 201-202.
The findings are presented appropriately and are sufficiently illustrated with participant voice. The discussion appears relevant to the findings presented.
Reviewer 2 Report
I read with interest the manuscript "The Impact of Neglected Tropical Diseases (NTDs) on Women’s Health and Wellbeing in sub-Saharan Africa (SSA) -A Case study of Kenya". Unfortunately, the authors did not give the depth that the theme needs in the results and it is not possible to understand how the text differs from everything that has already been published on the theme. In addition, many methodological flaws make this task even more difficult.
Next, I point out what makes me think that way.
Introduction
-It is extensive, repetitive and sometimes verbose. This section must be reduced by at least half. I suggest removing table 01 and diluting this information. Focus on explaining what NTD is, why women can have greater consequences for their well-being and why it is important to study this in Kenya.
Method
Line 108 to 120. I don't think that information should be here. I believe that it is more adequate in presenting the problem;
A series of information is needed to validate the method:
Which author / s conducted the interview or focus group?
What were the researcher’s credentials? E.g. PhD, MD
What was their occupation at the time of the study?
Was the researcher male or female?
What experience or training did the researcher have?
Was a relationship established prior to study commencement?
What characteristics were reported about the interviewer / facilitator? What methodological orientation was stated to underpin the study?
How were participants selected?
How were participants approached?
How many people refused to participate or dropped out? Reasons? Where was the data collected?
Was anyone else present besides the participants and researchers?
Were questions, prompts, guides provided by the authors? Was it pilot tested?
Were repeat interviews carried out? If yes, how many?
What was the duration of the interviews or focus group?
Was data saturation discussed?
Were transcripts returned to participants for comment and / or correction?
Methods and Results
Please, Promote a methodological description of what you did to arrive at the results, citing the approach used, how you defined the space and the universe of research; how the research group and the sample decided, if applicable; with which instruments did the data collection work operate; how you ordered, classified and analyzed the material. The results are still restricted to a brief categorization of the data based only on the reading of the statements and not on the conceptual strength of what is said.
Author Response
Reviewer 2
I read with interest the manuscript "The Impact of Neglected Tropical Diseases (NTDs) on Women’s Health and Wellbeing in sub-Saharan Africa (SSA) -A Case study of Kenya". Unfortunately, the authors did not give the depth that the theme needs in the results and it is not possible to understand how the text differs from everything that has already been published on the theme. In addition, many methodological flaws make this task even more difficult.
Next, I point out what makes me think that way.
- Introduction
It is extensive, repetitive and sometimes verbose. This section must be reduced by at least half. I suggest removing table 01 and diluting this information. Focus on explaining what NTD is, why women can have greater consequences for their well-being and why it is important to study this in Kenya.
We have shortened the introduction section and improved on table 01 by renaming it Major health issues caused by NTDs.
- Line 108 to 120. I don't think that information should be here. I believe that it is more adequate in presenting the problem.
We have moved the line 108-120 to the introduction section Line 101-106.
- A series of information is needed to validate the method:
We have expanded the data collection section and highlighted the corresponding paragraphs in the materials and methods section
- Which author / s conducted the interview or focus group? We have highlighted paragraph 176-177
- That were the researcher’s credentials? E.g., PhD, MD-line 133-134
- What was their occupation at the time of the study?-Line 133-134
- Was the researcher male or female?-Line 133-134
- What experience or training did the researcher have?Line 133-134
- Was a relationship established prior to study commencement? Line 135-138
- What characteristics were reported about the interviewer / facilitator? What methodological orientation was stated to underpin the study? Line 173-180
- How were participants selected? Line 129-131, 146-147
- How were participants approached? Line 135-149
- How many people refused to participate or dropped out? Reasons? Where was the data collected? Line 145-146
- Was anyone else present besides the participants and researchers? 155-156
- Were questions, prompts, guides provided by the authors? Was it pilot tested? Line 153
- Were repeat interviews carried out? If yes, how many? Line154-155, 162
- What was the duration of the interviews or focus group? Line 154, 162
- Was data saturation discussed? Line 154-155, line 162-163
- Were transcripts returned to participants for comment and / or correction? 183-186
Methods and Results
- Please, Promote a methodological description of what you did to arrive at the results, citing the approach used, how you defined the space and the universe of research; how the research group and the sample decided, if applicable; with which instruments did the data collection work operate; how you ordered, classified and analyzed the material. The results are still restricted to a brief categorization of the data based only on the reading of the statements and not on the conceptual strength of what is said. Line 190-204
Reviewer 3 Report
- Table 1 should be updated with more relevant information: the column name "Common symptoms" should be changed to "Major health issues".
- The major health issues due to Chagas disease is cardiomyopathy [Reference: Mitra and Mawson, 2017]
- The major health issues due to Buruli ulcer is necrotizing skin lesions.
- The symptoms and signs that occur due to Echinococcosis depend on the cyst's location and size.
- Dengue and Chikungunya are two diseases entities. They should be separated out.
- Symptoms of dengue fever depend on the types of the disease. Dengue haemorrhagic fever is a severe form of the disease, which causes platelet depletion and haemorrhage, sometimes leading to death.
- Chikungunia causes joint pain and joint swelling which may last for months or years. Fatality due to Chikungunia is 1 in 1,000.
- I just highlighted some examples. The authors must describe the major illnesses or health issues (not common symptoms) of all the NTDs, which will make people aware of the problem.
- Trachoma is a highly contagious disease. When narrating the focus group results on page 7, the authors should have mentioned the infectiousness of the bacteria Chlamydia trachomatis.
- Among the limitations, the authors should mention the importance of quantitative data, in addition to qualitative data analyses.
Author Response
- Table 1 should be updated with more relevant information: the column name "Common symptoms" should be changed to "Major health issues".
We have revised the entire table accordingly
- The major health issues due to Chagas disease is cardiomyopathy [Reference: Mitra and Mawson, 2017]
- The major health issues due to Buruli ulcer is necrotizing skin lesions.
- The symptoms and signs that occur due to Echinococcosis depend on the cyst's location and size.
- Dengue and Chikungunya are two diseases entities. They should be separated out.
- Symptoms of dengue fever depend on the types of the disease. Dengue haemorrhagic fever is a severe form of the disease, which causes platelet depletion and haemorrhage, sometimes leading to death.
- Chikungunia causes joint pain and joint swelling which may last for months or years. Fatality due to Chikungunia is 1 in 1,000.
- I just highlighted some examples. The authors must describe the major illnesses or health issues (not common symptoms) of all the NTDs, which will make people aware of the problem.
- Trachoma is a highly contagious disease. When narrating the focus group results on page 7, the authors should have mentioned the infectiousness of the bacteria Chlamydia trachomatis. Line 248-250
- Among the limitations, the authors should mention the importance of quantitative data, in addition to qualitative data analyses. Line 430-435
Round 2
Reviewer 2 Report
Thank you very much for reviewing the manuscript. In general, I do not believe that the authors have expended much effort to understand the suggestions or even improve the manuscript. An example is the suggestions that the authors say they incorporated in the introduction, although they only made a specific change in table 01.
After reading version1, the inaccuracies about the method, results and discussion still persisted. That is why it would be necessary to have new notes and a new reading in a process that can take a long time.
1. “Methodological transparency in qualitative research is a factor that contributes to its reliability, and must be guaranteed by researchers (Fontanella et al. 2011)” and ensured by the reviewers of a scientific journal, added as a reviewer.
According to Lincon and Denzin (2006), qualitative research has been changing over time, with the 1990s being a milestone in this process. The changes that strengthen and highlight the contribution of qualitative research, keep the problem of validity and reliability in focus, which is a constant challenge.
In this way, the basic that is expected from the authors, has not been fulfilled, for example to explain the theoretical saturation? Not only to say that it was made, but how it was made, how it was measured, by whom, using which methodological framework ... This simply does not appear.
“Closing the sample means defining the set that will support the analysis and interpretation of the data. In the non-probabilistic (intentional) samples, this definition is made from the researcher's experience in the research field, in an empiry based on reasoning instructed by theoretical knowledge of the relationship between the object of study and the corpus to be studied. If there was no closure due to exhaustion (addressing all eligible subjects), it must be justified why the processing of new observations and the recruitment of new participants were interrupted, but it makes clear the process by which this was done.
2. The analysis process and results should be described in minimum and sufficient detail so that readers have a clear understanding of how the analysis was performed, its strengths and limitations (Elo and Kynga¨s 2007 JAN).
3. The description of the data collection and analysis process includes a coming and going that confuses the reader. Despite describing the text, it lacks basic information that makes sense of the results presented. It is general information that is not linked to the studied content.
4. The results are still restricted to a repetition of speeches without making clear the system of analysis of the presentation. If there was a systematic it must be clear. In the presentation of the qualitative analysis, two paths are usually possible: describe the data categorized according to relevance presented by the interlocutors in the field, while discussing them in depth and comprehensiveness; or, first, make the description by means of classifications that unfold from categories brought from the field, and then produce a discussion that is based on the empirical, but shows the conceptual strength of what is said.
References
Graneheim U.H., Lundman B. Qualitative content analysis in nursing research: concepts, procedures and measures to achieve trustworthiness. Nurse Education Today, 2004. V.24, p. 105–112. doi: 10.1016 / j.nedt.2003.10.001
Elo S., Kynga¨s H. The qualitative content analysis process. J Adv Nurs. 2008. 62 (1): 107-15. doi: 10.1111 / j.1365-2648.2007.04569.x.
Fontanella B. J.B. et al. Sampling in qualitative research: proposal of procedures to verify theoretical saturation. Cad. Saúde Pública, 2011. V.27 (2): 389-394. doi: 10.1590 / S0102-311X2011000200020
Fontanella BJB, Ricas J, Turato ER. Saturation sampling in qualitative health research: theoretical contributions. Cad Saúde Pública 2008; 24: 17-27.
Denzin N.K. and Lincoln Y. The planning of qualitative research - Theory and Approaches. 2006. Editora ArtMed.
Author Response
We thank the reviewer for these comments, and we have revised the manuscript to address the above changes. Please see Line 130-148; Line 148-151; Line 156-161;Line 178-191; Line 194-209
Reviewer 3 Report
- Please note that the study results and discussion on the findings of your study should be presented in past tense.
- I suggest you to add the prevalence and a short description of pathophysiology of the NTDs in Table1. You can find those information from the following study: Mitra, A.K.; Mawson, A.R. Neglected tropical diseases: Epidemiology and global burden. Trop Med Infect Dis 2017, 2, 36, doi:10.3390/tropicalmed2030036.
- Minor: You can delete the serial numbers of the topics in the table.
Author Response
Please note that the study results and discussion on the findings of your study should be presented in past tense.
We have revised the results to past tense in the discussion section.
I suggest you to add the prevalence and a short description of pathophysiology of the NTDs in Table1. You can find those information from the following study:
Mitra, A.K.; Mawson, A.R. Neglected tropical diseases: Epidemiology and global burden. Trop Med Infect Dis 2017, 2, 36, doi:10.3390/tropicalmed2030036.
We take the reviewer’s point that this information would be very useful for the interested reader. However, we do not want to distract from the main points of the manuscript. We have therefore pointed the reader to this reference for more information. (then do that…)
Minor: You can delete the serial numbers of the topics in the table.
Revised to remove the serial numbers.